# A Systematic Review on Attachment and Sleep at Preschool Age

**DOI:** 10.3390/children8100895

**Published:** 2021-10-07

**Authors:** Catarina Perpétuo, Eva Diniz, Manuela Veríssimo

**Affiliations:** William James Center for Research, ISPA—Instituto Universitário, 1100-304 Lisbon, Portugal; cperpetuo@ispa.pt (C.P.); EDiniz@ispa.pt (E.D.)

**Keywords:** attachment, sleep, actigraphy, parental reports, preschool age

## Abstract

Sleep is a biological process that impacts nearly every domain of a child’s life. Sleep-wake regulation influences and it is highly influenced by developmental variables related to parent-child relationships, such as attachment. The main goal of the present systematic review is to analyze and integrate the findings of empirical studies investigating the relations between attachment and sleep in preschool age, a period marked by important developmental changes that challenge both attachment system and sleep-wake regulation. A database search was performed using a combination of relevant keywords, leading to the identification of 524 articles, with 19 manuscripts assessed for eligibility; finally, seven studies (2344 children) were included. Overall, the findings were not consistent, with some studies reporting significant associations between attachment security and sleep quality, as well as between attachment insecurity and sleep problems, whereas others did not find significant associations. The results are discussed in light of the available theoretical models and integrated in the context of measurement approaches to attachment and sleep heterogeneity, aiming to guide future research on the topic.

## 1. Introduction

Sleep is a relationally guided biological process, with implications in nearly every domain of child development, holding a core place in every child’s life. During early infancy and preschool age, children spend approximately half of their time asleep [1,2,3], suggesting that sleep may be particularly important during periods of greater plasticity, playing a role in neuronal maturation and development [4,5,6]. However, the influence of sleep in child development goes far beyond biological processes, impacting the child’s health and well-being [7], cognitive and behavioral functioning [8], affect regulation [9], and social competence [10,11,12].

Despite the importance of sleep in multiple domains of child development, sleep disorders are highly common, occurring in 20 to 50% of children at some point of their development [13,14] and persisting through subsequent years in 8.5 to 12% of the cases [15,16]. Sleep problems may uncover a health issue, considering their associations with negative physical [16] and mental health outcomes [17], pointing to bidirectional links between sleep and health variables. Sleep problems also relate to greater internalizing and externalizing behavioral problems, both concurrently and longitudinally [18,19,20], social-emotional problems [21,22], and poorer executive functioning [23]. Given the high prevalence of sleep problems during early childhood and its detrimental impact in child development, efforts have been made in order to uncover the factors that influence child’s sleep. As such, Sadeh and Anders [24] theorized the transactional model of sleep-wake regulation [24,25,26,27], conceptualizing how sleep both influences and is influenced by the characteristics of child’s developmental contexts, such as culture, environment, and family relationships. According to this model, infant sleep regulation has a dynamic bidirectional relation with: (a) distal determinants that shape the parent context, affecting infant’s sleep secondarily (e.g., cultural and socioeconomic influences); (b) intermediate factors (e.g., parental health, personality, cognitions, and expectations about childhood and sleep); and (c) proximal influences (e.g., infant biomedical and constitutional factors, such as temperament and physiological reactivity, as well as specific parenting interactions) [24]. The authors also emphasize the role of child-parent dyadic relationships, particularly attachment, mediating the relations between sleep, and the referred contexts during the first years of life. However, given the lack of associations between attachment and sleep before 2 years of age [28], possibly due to a high intra-individual variability in sleep patterns, we decided to focus this review on preschool children (2 to 5 years old).

As going to bed constitutes a separation between child and parents, which is frequently experienced with anxiety, and one of the functions of the attachment system is to provide security to the dyad in times of distress, sleep is proposed to act as a trigger to the attachment system. Given the importance of attachment relationships on sleep regulation, the current review will focus on the interpersonal system of the transactional model of sleep regulation. Some theoretical explanations about the connection between attachment and sleep, derived from this model, will be detailed next. 

### Attachment and Sleep

Attachment relationship is conceptualized as the enduring and stable emotional bond between the child and the caregiver, ensuring the caregiver’s proximity in threatening or stressful situations to provide physical and psychological security to the child [29,30,31]. As the child experiences his/her caregivers as warm, sensitive, and consistent in meeting his needs, or rather as cold, unsupportive, and unpredictable, a secure or insecure attachment is formed correspondingly. Secure attachment bonds relate to better developmental outcomes, such as higher rates of executive functioning [32,33,34], language development [35], emotional understanding [36], social competence [37,38], and resilience [39]. In turn, insecure attachment relationships are related with less optimal developmental outcomes, such as increased risk of developing disease [40], delayed development of discourse [41], and socio-emotional difficulties, measured both cross-sectionally and longitudinally [42,43,44].

The attachment system and sleep-wake regulation have been subject to a considerable amount of theoretical and empirical work that has been trying to uncover how these systems may relate [9,24,25,45,46,47]. Some conceptual and theoretical explanations for the relations between sleep and attachment have been advanced and grouped into four distinct models [47]. The first one suggests that behaviors related to sleep problems are attachment behaviors in reaction to night-time separations [24,25,48]. In light of this model, bedtime resistance could be an expression of separation anxiety and night-wakings would work as a mechanism of proximity seeking towards the attachment figure. The second model theorizes sleep problems as a consequence of attachment insecurity. Fears and worries related to bedtime separation can compromise a child’s capacity to relax and downregulate, physiologically and psychologically, in order to fall asleep [9,46]. In opposition, the third model conceptualizes sleep problems as a source of insecure attachment. Because sleep problems impact children’s emotional regulation [49], it may affect the formation of social bonds [45], posing the dyad at a greater risk of developing an insecure attachment. Finally, the last model hypothesizes that inconsistent parenting may be the cause of the relations between sleep and attachment [46].

Empirical evidence, however, has not always been consistent in supporting the outlined models. Studies examining sleep problems and attachment are still sparse and insufficient, generating results that are often difficult to compare, namely by assessing distinct age ranges, or using different instruments to measure different domains of attachment or sleep. For example, studies evaluating the behavioral quality of attachment (e.g., Strange Situation Procedure [50]; Adaptation for preschool-age [51]; Attachment Q-Sort, [52]), or its representation (e.g., Attachment Story Completion Task, ASCT, [53]), may lead to distinct results [54]. This is particularly significant considering the moderate correlations found between different attachment measures [55,56], suggesting some overlap between measures, but also revealing that different measures may assess distinct aspects of attachment. 

Similar challenges are faced in sleep assessment, with different findings regarding objective (e.g., actigraphy) and subjective measures (e.g., parent’s self-report). On the one hand, objective recordings tend to overestimate night-wakings and to miss important behavioral aspects of sleep phenomena, such as bedtime resistance and sleep anxiety [56,57,58,59]. On the other hand, parental reports tend to overrate sleep duration and wake-up time, underestimating bedtime/night-wakings [60,61].

Despite the inconsistence of the research findings between attachment and sleep during infancy, the relations between them across the lifespan have been described in a systematic review [62], suggesting the existence of bidirectional relations between sleep and attachment, starting early in infancy (0 to 2 years old) and continuing throughout life, in children (2 to 18 years old), adults (18 to 64 years old), and seniors (>65 years old). One meta-analysis also examined children’s sleep (between 6 months and preschool age; [28]), describing positive associations between some parameters of sleep and attachment security. Specifically, attachment security correlated with higher sleep efficiency and lower rates of sleep problems, while attachment resistance correlated positively with maternal related sleep problems. Despite the importance of these findings, this meta-analysis focused on the comparison between secure and resistant attachment styles, failing to consider conclusions regarding other attachment styles (i.e., avoidant and disorganized). Moreover, it also neglected the studies that adopt continuous measures of attachment security (i.e., Attachment Q-Sort (AQS) and Attachment Story Completion Task (ASCT)), leaving it understudied how attachment may be related with sleep.

Aiming to overcome some of the limitations of previous reviews (e.g., wide age-range, inconsistent findings), in the current review, we focused on a specific developmental period: preschool age, i.e., between 2 and 5 years old. This age range was defined given that although sleep consolidation happens at around 12 months (i.e., the distribution of sleep in one longer period during the night and a short nap during the day), due to some individual variability in this process, sleep consolidation does not tend to be considered until two years old [63,64,65]. Moreover, preschool age is a period when developmental processes (e.g., the transition to day-care and the redefinition of the parent-child relationship into a goal-corrected partnership) may challenge both the attachment system and sleep-wake regulation [50,66] or the constitution of new roles and responsibilities [67], hampering the relations between sleep quality and attachment at early stages of development [12,68].

Bearing that in mind, as well as the specific contributions of previous reviews, the current review aims to provide a systematized and integrated overview of studies on sleep quality and attachment relationships in preschool children (i.e., age range between 2 and 5 years old). Specifically, it will examine how objective and subjective sleep dimensions may be related to different dimensions of attachment, aiming to illuminate directions for future research.

## 2. Materials and Methods

The present systematic review follows the PRISMA (*Preferred Reporting Items for Systematic Reviews and Meta-Analyses*; [69]) general guidelines for reporting data concerning the relations between attachment and sleep in children aged between 2 and 5 years old. Every step of the process of identification and selection of articles is specified below.

### 2.1. Information Resources and Search Strategy

A systematic data search was performed using the following string: sleep AND attachment AND (child* OR toddler* OR preschool* OR years OR months), on titles, abstracts, and keywords, in the following databases: EBSCO (PsycInfo, PsycArticles, Academic Search Complete, and Psychology and Behavioral Sciences Collection) and Web of Science. The search covered the period until 21 September 2020 (no starting date limit), resulting in 524 titles. The initial abstracts, identified as potentially relevant articles, were screened by both authors separately, and a total of 505 papers were excluded at this phase. The remaining 19 articles were fully assessed independently, and after disagreements were resolved by consensus, a total of seven articles were included (Table 1).

### 2.2. Eligibility Criteria, Information Resources and Search Strategy

Articles were included in this review according to the following inclusion criteria: (1) empirical studies with an available abstract, published in peer-reviewed journals; (2) articles written in English, Portuguese, French, or Spanish (languages mastered by the authors); and (3) articles assessing children’s sleep between 2 and 5 years old. 

We excluded all the studies in which: (1) children or parents were not living in natural contexts (e.g., institutionalized or hospitalized children, incarcerated parents, etc.); (2) children were premature or diagnosed with some physical and/or mental illness; (3) the mother or father were diagnosed with some physical and/or mental illness; (4) there was not a specific aim to examine the relations between child’s attachment and sleep; (5) intervention programs or clinical trials were conducted; (6) there were no specific measures of child attachment and/or sleep. The flow diagram for the identification, screening, eligibility and inclusion of studies can be found in Figure 1.

For the purpose of this review, we included studies in which child’s sleep was assessed via actigraphy or sleep diaries for sleep parameters related to sleep quantity and quality, and parental-reported questionnaires for sleep problems and sleep-related behaviors. Given the amount of longitudinal designs found, we excluded studies that evaluated sleep exclusively before 24 months of age, given that, during the first two years of life, children display great intraindividual variability due to maturational processes [77].

### 2.3. Data Extraction and Items

A data extraction matrix was developed to summarize collected data, identifying: (1) general information about the participants, such as age range, number of participants, country of origin, socioeconomic status, and ethnicity; (2) study design; (3) attachment measure; (4) sleep measure; (5) main findings regarding the relations between attachment and sleep. Data from the full papers were extracted by both researchers independently. 

## 3. Results

### 3.1. General Description of the Studies: Theoretical Perspectives

Most of the included studies are rooted in the perspective that sleep difficulties may partially result from vicissitudes in the attachment relationship ([9]; ID#1, 3, 4, 5). This framework stands that insecurely attached children may experience more intense fears and worries, which might disrupt sleep. In one case, both frameworks of attachment as a predictor of sleep [9] and sleep as predictor of attachment [44] were considered (ID#2). One study (ID#6) drew upon a conceptual framework considering sleep and attachment as dual products of parenting [46]. Finally, one of the studies did not specify the authors’ standpoint regarding associations between sleep and attachment (ID#7).

### 3.2. General Sociodemographic Characteristics of the Participants

All the included studies were conducted in North American countries (four in Canada: ID#1, 2, 3, 4, and three in the USA: ID#5, 6, 7), and the majority of the children were Caucasian (80% across all studies; one study did not provide information regarding ethnicity). A large proportion of the children were integrated into intact families, meaning that 89% lived with both parents in the same household (two studies did not specify parental marital status; ID#3, 6). Overall, children belonged to well-educated families, with 81% of the mothers having completed high school and at least 76% holding college degrees (two studies did not provide information on maternal education; #3, 6).

### 3.3. Study Designs

Most of the studies relied on longitudinal designs, either with attachment and sleep measured at different ages, or with sleep assessments across distinct time points (ID#1, 2, 3, 4, 5, 7), whereas only one study was conceived with a cross-sectional design (ID#6). Although some studies reported sleep assessments before the children were 2 years old (ID#1, 2, 3, 8), these data were not considered in this systematic review. As such, we analyzed longitudinal data from four studies (ID#1, 3, 4, 7), cross-sectional data from two studies (ID#2, 6), and both longitudinal and cross-sectional data from one study (ID#5).

### 3.4. Empirical Processes on Sleep and Attachment Measures

Most of the articles examined sleep as an outcome of the attachment relationship, whereas only one investigated attachment as an outcome of sleep (ID#2). Moreover, mechanisms influencing the relations between sleep and attachment were examined only by one study (ID#5), uncovering the role of a temperamental variable, i.e., negative emotionality. Finally, one study (ID#7) identified how individual (e.g., child’s sex, temperament) and mother-related characteristics (e.g., sensitivity, depression) accounted for developmental changes in the relations between attachment and sleep. Details on how attachment and sleep have been conceptualized and measured by the included studies will be specified next, followed by a description of the main findings of the associations between sleep and attachment.

### 3.5. Sleep and Attachment Measures across Studies

Sleep has been described as a complex, multifaceted, and multidimensional phenomenon, which can be studied objectively or via parental report in young children. Sleep assessment may involve different parameters, such as sleep quantity, sleep quality, and sleep-related behaviors. Table 2 illustrates the type of sleep instruments (i.e., actigraphy, sleep diary, questionnaire) used to assess sleep quantity, quality, or sleep problems.

Parameters of sleep quantity and/or quality were the focus of most of the identified research. Sleep parameters of quantity, i.e., sleep duration across a 24 h period, or estimates of it (e.g., bedtime, wake-up time), were one of the most frequent aspects examined (ID#1, 2, 3, 4, 6). Sleep quality variables, e.g., sleep efficiency, defined by the percent of time in bed spent effectively asleep and number of night-wakings, were also often assessed (ID#1, 3, 4, 6, 7). Aspects related to sleep fragmentation (e.g., frequency and length of night-wakings, physical agitation) were the most common parameters of sleep quality examined.

The assessment of sleep quality and quantity tended to rely on both actigraphy, i.e., an objective measure of sleep, and sleep diaries (ID#1, 2, 4). However, one study used only actigraphy (ID#6), and three studies exclusively relied on maternal reported questionnaires (ID#3, 5, 7).

Concerning attachment evaluation, the Strange Situation Paradigm [50] and its modified version suited for preschool age (Preschool Separation-Reunion Procedure, [51]) were frequent procedures to evaluate the attachment of the child to mother/parents (ID#3, 4, 7). Other studies relied on Attachment Q-Sort ([52]; #1, 2, 5) and one assessed attachment representations with the Attachment Story Completion Task ([53]; #6).

### 3.6. Associations between Attachment and Parameters of Sleep

Overall, studies examining attachment and quantitative parameters of sleep found inconsistent results. Table 3 summarizes the main findings of the primary studies. On the one hand, more securely attached children (at 15 months) slept more at night (at 24 months) than their insecurely attached counterparts, according to actigraphy (ID#1). Moreover, disorganized attachment was related to maternal perceptions of later bedtimes and shorter total sleep durations for their children (ID#3). On the other hand, some studies did not find associations between attachment and sleep parameters such as sleep duration, longest period of uninterrupted sleep, time in bed, or wake-up time (ID#2, 3, 4).

Regarding the quality of sleep, children classified as disorganized tended to be described by the mothers as having more fragmented sleep, i.e., more frequent and longer night-wakings, in comparison to children with other attachment classifications (i.e., secure and insecure-ambivalent; ID#3). More resistant attachment behaviors were also related to mothers’ reports of longer night-waking durations for children at 24 months (ID#4).

In turn, associations between attachment security and objectively measured sleep quality were inconsistent. On the one hand, secure attachment correlated positively with sleep efficiency (ID#1, 6) and negatively with night-waking duration and frequency, as well as with other variables indicative of less sleep quality, such as sleep activity (ID#6). On the other hand, these associations were not found in another study also relying on objective assessments of sleep (ID#4).

### 3.7. Associations between Atachment and Sleep Problems

The relations between attachment and sleep problems were only examined by two articles, again with inconsistent findings. Negative correlations between attachment and sleep problems were found by one study (ID#5), both cross-sectionally (at 24 months) and longitudinally (at 36 months). Furthermore, one study included a temperamental variable as a moderator of the relation between attachment and sleep, and found that the child’s negative emotionality moderated this association. Attachment security predicted fewer sleep problems for children with high negative emotionality, but not for children with low negative emotionality at 54 months (ID#5). However, another study did not find associations between attachment and sleep problems (ID#7), suggesting that the role of child variables, such as breastfeeding duration and temperament, and of maternal variables, such as maternal sensitivity and depression, impact developmental changes in sleep.

## 4. Discussion

The main aim of the present systematic review was to summarize research findings concerning the relations between attachment and sleep in children from 2 to 5 years old. A handful of studies suggest that children with secure attachment relationships have better overall sleep than insecurely attached children. For example, it has been reported that children with insecure (ID#1, 2) and with disorganized (ID#3) attachment relationships sleep for fewer hours (ID#1, 3), and, moreover, wakeup more frequently during sleep and stay awake for longer periods (ID#2, 3) than securely attached children, who in turn sleep more efficiently (ID#1, 6).

One possible explanation for these results involves attachment and self-regulation theories [78]. Accordingly, secure attachment relationships are built upon a history of repeated experiences of successful dyadic regulation [28,29,79], resulting in the development of a stronger capacity for physiological [80] and emotional self-regulation [81]. This capacity can be manifested at bedtime, with securely attached children being more able to peacefully self-regulate in order to fall asleep and to settle after a night-waking [9], increasing total sleep time and sleep efficiency. On the contrary, children with insecure/insecure-ambivalent/disorganized attachment classifications would experience a more difficult time when they face the transition to sleep alone in the dark [47,82]. Darkness and loneliness are natural clues to danger, defined as stimuli that, although not inherently dangerous, increase the likelihood of danger, leading to the activation of the attachment system [29]. Insecurely attached children are more prone to intensely experience, and express, separation anxiety than secure children [83]. Accordingly, their emotional reactions when facing night-time separations can increase bedtime resistance, delaying bedtime and shortening total sleep time and sleep efficiency [84]. Concomitantly, a large proportion of insecure children would be more likely to signal their night-wakings (ID#4) as part of a mechanism to seek proximity from caregivers, extending their durations and making their parents more likely to notice and report sleep problems in retrospective questionnaires [85]. This explanation is reinforced by the findings showing that mothers of insecure-ambivalent (ID#4) and disorganized (ID#3) children report more frequent and longer night-wakings than mothers of securely attached children. These findings may explain the abovementioned associations between insecure attachment and sleep.

Another possible explanation is that parents of securely attached children may implement more stable bedtime routines, which, in turn, lead to better sleep [64,86,87]. Although, to our knowledge, no study has investigated the relations between attachment security and bedtime routines, those have been associated with variables concerning parent-child relationships [88,89,90,91] and parenting styles [92], suggesting that the perception of stability originating or reinforced by routines may improve family relationships. However, adopting the framework of attachment theory, it has been reported that parents of securely attached children structure their environment, making it predictable and stable. This capacity had been described as a part of sensitive caregiving that characterizes parents of securely attached children.

However, other studies did not find significant associations between sleep and attachment (ID#2, 4, 7), especially when sleep parameters, such as number and length of night-wakings, were measured objectively (ID#2, 4). This lack of associations suggests that attachment may be linked to sleep dimensions that are easily assessed by maternal reports, such as night-time and sleep-related behaviors, but not through objective measures, such as actigraphy [73]. Hence, different instruments capture distinct dimensions of sleep phenomena. While objective methods (e.g., actigraphy) assess objective sleep duration, timing, and quality (i.e., sleep efficiency) and tend to overestimate night-wakings [57,93,94], subjective methods (e.g., sleep diaries and questionnaires) reflect maternal perceptions of the child’s sleep that are more limited as the child gets older [95]. In that sense, some authors have suggested that relevant sleep parameters must be selected based on developmental considerations [96]. As such, variables like the proportion of night-time sleep and longest period of uninterrupted sleep, which tend to stabilize over time [63], should be considered. These findings point to the relevance of measuring sleep both via maternal reports and objective methods [85], assessing developmental relevant variables, in order to account for behavioral and objective dimensions of sleep that may be related to attachment.

The reported inconsistency among the findings may mirror not only the variability of sleep parameters assessed, but also the diversity of attachment dimensions measured across the studies. Attachment was measured in different contexts (i.e., laboratory ID#3, 4, 6, 7, and home ID#1, 2, 5) and via distinct coding systems (i.e., typological ID#3, 4, 7, and continuous ID#1, 2, 5, 6). There is evidence that typological approaches used to score attachment behaviors in the Strange Situation Paradigm may not be the best assessing for the variability in the quality of mother-child attachment relationships. Otherwise, instruments yielding continuous scores of security seem to capture more dynamic aspects of the attachment relationship [97] and show excellent variation in low-risk samples [73]. Moreover, it is possible that different components of the attachment relationship contribute differently to the child dealing with bedtime separations, impacting distinct sleep parameters. Some studies found that secure attachment, assessed by the AQS [52,98], was associated with higher sleep duration and efficiency (ID#1) and fewer sleep problems (ID#5). It is plausible that the aspects of a child’s secure base behavior, measured in a low-stress, known home environment [52,98], are those that help him/her to bear bedtime separations from the caregiver and to fall asleep. Conversely, studies that assessed attachment separation-reunion behaviors through the Strange Situation Paradigm [50] reported interesting findings regarding disorganized (ID#3) and ambivalent (ID#4) classifications. Both tended to wake up more frequently and for more minutes during the night, and mothers of disorganized children reported shorter sleep durations. Given that bedtime separations and morning or middle-of-the-night reunions are an ecological repetition of the Strange Situation Paradigm, it is credible to think that children who report high levels of separation stress, and cannot calm down easily after reunion, live analogous experiences every day at bedtime.

Although according to the transactional model of sleep-wake regulation [25,26,27,82], parent-child relationships are a major factor impacting child’s sleep, linking directly with attachment theory, and research is still sparse and insufficient. As reported in the present review, no study has investigated attachment and sleep at preschool age since 2015 and all of the studies were conducted in North American countries. This, along with the fact that nearly all of the children who participated in the primary studies belonged to well-educated, medium-high socio-economic status samples, may compromise generalization. It is plausible that attachment quality, as much as sleep parameters, are less likely to differ significantly in such samples, therefore making effects harder to detect. Nevertheless, socio-economic status has been reported to affect both attachment security [99,100,101] and child’s sleep [102,103,104].

Another aspect worth mentioning concerns two assumptions regarding the attachment relationship that should be analyzed carefully: stability across time and stability across distinct relationships. First, the fact that all but one study has a longitudinal design is noteworthy, expressing an effort to uncover how attachment is related to the development of sleep. However, although there is a tendency for attachment stability across the lifespan [105,106], it is also known that attachment classifications can change in the early years, particularly when assessed via behavioral and representational measures [107]. The absence of control for attachment stability possibly weakens the studied effects. Second, all of the studies that assessed attachment behaviors (ID#1, 2, 3, 4, 5, 7) evaluated only mother-child dyads, despite the evidence pointing to modest to moderate correlations between mother-child and father-child attachment [107,108]. Knowing that the presence of at least one secure attachment relationship to one of the parents buffers the negative impacts of insecure attachment to the other parent [109,110], evaluating attachment to one single figure limits the understanding of attachment-sleep relations. Apart from the concordance between child-mother and child-father attachment relationships, it has been established that father-child reciprocal interactions have a unique contribution to child development [111,112,113], for example, in behavioral regulation [114]. The importance of father-related variables [115], such as father’s mental health [116,117,118], paternal emotional support [119], and particularly paternal involvement in child sleep, have been demonstrated in infants [120], preschool-aged [121], and school-aged children [122]. However, to our knowledge, none of the studies that investigated the relations between attachment and sleep at preschool age included measures of father-child attachment, undermining the understanding of father-child attachment influence on sleep.

Despite the relevance of the findings, some limitations of the current review should be addressed. First, the number of primary studies was relatively small, with most of identified studies collecting data from small samples (ID#1, 2, 4, 6). Additionally, most of the children were from middle class western societies and belonged to well-educated families, compromising the generalization of results due to the lack of sample variability and representatively. Secondly, by only including empirical scientific articles, potentially relevant sources of information about the topic published elsewhere (e.g., grey literature, book chapters) were not identified. Third, the use broad keywords during the article search, such as “sleep,” “attachment,” or “child,” may have left out articles related to the research topic indexed with other terms. Lastly, included studies that used objective sleep measures tended to focus on short spans of time (i.e., 72 h, #1, 2, 4), although the recommended duration for the use of actigraphy is at least seven consecutive days [123]. Thus, these findings should be examined carefully.

Paths for future investigation that have been uncovered in the current section are systematized below. First, selected samples should become more heterogeneous regarding sociodemographic and family factors, allowing researchers to study if the association between attachment and sleep is moderated by factors that were not included in the primary studies. Despite the evidence suggestive of the role played by factors such as maternal depression [124,125], maternal anxiety [126], or family socioeconomic status both in attachment security [99] and child’s sleep [103], these were overlooked by the included studies, ignoring the influence of these factors in the relations between attachment and sleep. Second, attachment assessments should privilege instruments that yield continuous scores of attachment, both to the mother and father or other primary caregiver, and consider the possibility of a second assessment if it was first measured before 18 months old. Finally, future studies should focus on capturing sleep phenomena in all their amplitude, using both objective and subjective sleep measures over extended periods, and favoring the assessment of developmental relevant variables.

## Figures and Tables

**Figure 1 children-08-00895-f001:**
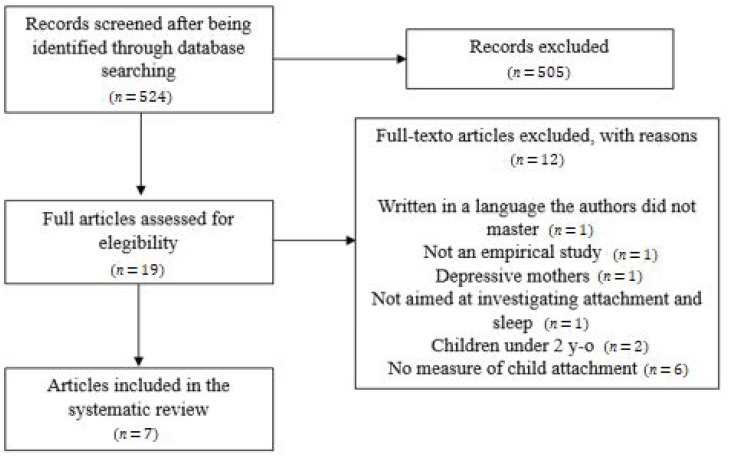
PRISMA flow diagram for the identification, screening, eligibility, and inclusion of studies.

**Table 1 children-08-00895-t001:** References of the primary studies.

ID	Reference	Participants	Design
#1	Bélanger, Bernier, Simard, Bordeleau, & Carrier (2015) [70]	*N* = 62 (30 girls); mostly Caucasian; Medium-High SES	Longitudinal
#2	Bernier, Bélanger, Tarabulsy, Simard, & Carrier (2014) [71]	*N* = 63 (27 girls); mostly Caucasian; Medium-High SES; intact families.	Longitudinal
#3	Pennestri et al. (2015) [72]	*N* = 134 (71 girls); mostly Caucasian; Medium-High SES	Longitudinal
#4	Simard, Bernier, Bélanger, & Carrier (2013) [73]	*N* = 55 (25 girls); mostly Caucasian; Medium-High SES; intact families	Longitudinal
#5	Troxel, Trentacosta, Forbes, & Campbell (2013) [74]	*N* = 776 (393 girls); mostly Caucasian; Parental education 14 years average	Longitudinal
#6	Vaughn et al. (2011) [75]	*N* = 39 (12 girls); 36% from ethnic minorities, mostly AA	Cross-sectional
#7	Weinraub et al. (2012) [76]	*N* = 1215. Maternal education 14.3 years average; 13.3% single mothers; 31% low income families	Longitudinal

SES stands for Socioeconomic Status, and AA stands for African American.

**Table 2 children-08-00895-t002:** Type of sleep instruments and variables measured across the studies.

ID	*N*	Attachment Variables and Instruments	Age at Sleep Assessment	Measures		
				Sleep Quantity	Sleep Quality	Sleep Problems
#1	62	Attachment Q-Sort (AQS): security and independence	(15 m), 24 m	Actigraphy, sleep diary: sleep minutes at night and over the 24-h period	Actigraphy: sleep efficiency	-
#2	63	AQS: security	(12 m), 24 m	Actigraphy, sleep diary: night-time sleep duration and proportion of night time sleep to total sleep	-	-
#3	134	Preschool Separation-Reunion Procedure: attachment classification (secure, ambivalent, avoidant, disorganized)	(6 m, 12 m), 24 m, 36 m	Adaptation of Self-Administered Questionnaire for the Mother: bedtime, wake time, sleep latency, nocturnal sleep duration, longest period of uninterrupted sleep	Adaptation of SAQM: nr of night wakings	Adaptation of SAQM
#4	55	Strange Situation Procedure (SSP): proximity seeking, contact maintenance, avoidance, resistance	24 m	Actigraphy, sleep diary: sleep duration at night, wake duration at night	Actigraphy: nr of nocturnal awakenings	-
#5	776	AQS: security	24 m, 36 m	-	-	Sleep problems subscale of the CBCL
#6	39	Attachment Story Completion Task (SSP): coherence and security	4-5 y	Actigraphy: sleep duration, total sleep minutes, longest wake episode, sleep latency	Actigraphy: wakings after sleep onset, sleep activity mean, overall activity index, sleep efficiency	
#7	1215	SSP: security; attachment classification; Separation distress	(6 m, 15 m), 24 m, 36 m		Maternal interview: night wakings in the previous week	Sleep problems subscale from CBCL

The ages placed between brackets refer to ages assessed by primary studies that were excluded from the current review. SAQM stands for Self-Administered Questionnaire for the Mother and CBCL stands for Child Behavior Checklist.

**Table 3 children-08-00895-t003:** Main findings of the primary studies.

ID	Reference	Main Findings
#1	Bélanger, Bernier, Simard, Bordeleau, & Carrier (2015) [70]	Positive associations between security and: sleep minutes at night and sleep efficiency. Negative associations between dependency and sleep minutes at night.
#2	Bernier, Bélanger, Tarabulsy, Simard, & Carrier (2014) [71]	Attachment security was unrelated to night-time sleep duration and to proportion of night-time sleep to total sleep.Insecure attachment was related to night-wakings.
#3	Pennestri et al. (2015) [72]	Disorganized attachment associated with a shorter duration of nocturnal sleep than secure or ambivalent attachment; more night awakenings; shorter periods of uninterrupted sleep.
#4	Simard, Bernier, Bélanger, & Carrier (2013) [73]	Ambivalent attachment associated with more minutes awake at night as perceived by their mothers, but not according to actigraphy records.
#5	Troxel, Trentacosta, Forbes, & Campbell (2013) [74]	Secure attachment correlated with fewer sleep problems. No evidence was found for a statistically significant direct path between attachment security and toddler sleep problems at 36 months. Significant direct relations were found between attachment security and sleep problems, but only among children characterized with high negative emotionality at 6 months.
#6	Vaughn et al. (2011) [75]	Attachment was positively associated with sleep duration, Total sleep minutes and sleep efficiency; negatively associated with variables reflecting poor sleep quality (number of sleep episodes, sleep activity index, wake minutes after sleep onset).
#7	Weinraub et al. (2012) [76]	Infant-mother attachment was not related to sleep problems.

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
