# Peer review of "A Systematic Review on Attachment and Sleep at Preschool Age"

_children, 2021, doi:10.3390/children8100895_

Round 1

Reviewer 1 Report

This is an interesting scoping review that includes studies aimed at investigating the relationship between sleep in preschool-aged children and attachment quality.

Although the interesting focus of this paper, the manuscript is in need of some modification to be ready for publication. The following comments are meant to be constructive in the possibility of a revision of the manuscript.  

  • Page 1, line 40. If the transactional model is the theoretical framework that guides this review, it could be helpful for readers to briefer introduce all the factors that could influence sleep, according to the model, including intrinsic factors. Moreover, the parent-child interaction context should be described more in-depth, underlining that the review will be focused on the interpersonal system in terms of attachment.
  • Page 3, line 101. It is not clear what does it means that the cited metanalysis “only focusing on a dichotomic attachment perspective”. Could authors be more specific in described this work? Moreover, it is not clearly outlined what the current work adds to the metanalysis findings.
  • A more comprehensive table with all the information extracted will be more useful for readers. The table should be contained all the information on page 4 line 157 and information on table 2 (it is ok if authors prefer to maintain results in a different table). Moreover, I suggest inserting in the table a column for the attachment variables analyzed in each study (e.g., secure attachment, avoidant attachment, etc..) in addition to the attachment measures, and a column for the sleep variables analyzed in each study (e.g., total sleep duration, number of night awakenings) in addition to the sleep measures.
  • Moreover, considering that the authors commented in the results section on the presence of other measured variables in the analyzed studies (e.g., moderators) a column of the table should include this information.
  • Page 4, line 162. I was not able to see Fig. 1 (the flow chart). It was not in the supplementary materials.
  • Page 9, line 359. Authors should stress the importance of including fathers in sleep studies. Not only because mother-child attachment and fathers-child attachment are correlated but also because of the empirical findings regarding the important role that fathers play in children’s sleep development (e.g., Bernier, A., Tétreault, É., Bélanger, M. È., & Carrier, J. (2017). Paternal involvement and child sleep: A look beyond infancy. International Journal of Behavioral Development41(6), 714-722.; Tikotzky, L., Sadeh, A., & Glickman-Gavrieli, T. (2010). Infant Sleep and Paternal Involvement in Infant Caregiving During the First 6 Months of Life. Journal of Pediatric Psychology, 36(1), 36–46.; Ragni, B., De Stasio, S., & Barni, D. (2020). Fathers and sleep: a systematic literature review of bidirectional links between paternal factors and children's sleep in the first three years of life. Clinical Neuropsychiatry17(6))

Author Response

We genuinely thank the reviewers for their insightful and constructive feedback on our manuscript (1372252). The utmost attention was devoted to all of the comments. We have replied to all your comments/suggestions, aiming to clarify our conceptual background, methodological procedures, and data analyses.

A point-by-point reply now follows. All substantive changes in the manuscript were highlighted in yellow. We greatly appreciate the opportunity to improve the manuscript based on these comments and suggestions. We are confident that the comments and the ensuing revisions made the manuscript a stronger contribution to the field.

  1. We thank for the comment and the opportunity to clarify the Transactional Model. We now present the model with more details and clarify the focus of the review on the interpersonal dimension of the model, particularly the attachment system (please see p. 1, lines 43-61).
  2. We thank the reviewer by this important aspect. We now clarified the results of the cited meta-analysis, as well as its limitations, to outline what the current work adds to previous reviews on the topic.  This can be found between lines 117-124.

  3. We thank the reviewer this comment, allowing to better communicate the methods and findings of included studies. We reviewed Table 1 according to your suggestions.

    We also added a column on Table 2 detailing not only the attachment measure, but also the attachment variables being measured; sleep instruments and variables were also included. However, we did not detail moderators/mediators in Table 2, because only one studied use them. This information is described in Results’ section (please see p., lines 276-180).

  4. We apologize for that. Figure 1 was now included in supplementary variables.

  5. We thank the reviewer His/her insights about the importance of father involvement in children’s sleep. We fully agree with this point of view and we dedicated some more thoughts to the importance of father-child attachment in sleep and cited the suggested references, as well as some others (please see p., lines 393-402).

Reviewer 2 Report

Overall, the manuscript is well written and presents a methodologically sounds systematic review of the topic of sleep and attachment in preschool aged children.

Below are a few suggestions.

Introduction

  • Line 42- specify the authors as authors of the transactional model for clarity. For example, “The authors of the transactional model also emphasize…”
  • Line 59 (paragraph)- may be helpful to include the ages of the children in this attachment system since the study is specific to preschool age to establish the need for the current review.
  • Line 108- provide a definition of sleep consolidation as it relates to this study

Materials and Methods

  • Line 142- Justify the inclusion of only quantitative studies rather than mixed method or qualitative

Results

  • Table 2- Add in a column to describe the method of assessment of attachment used in the study or add in a separate table to describe

Author Response

We genuinely thank the reviewers for their insightful and constructive feedback on our manuscript (1372252). The utmost attention was devoted to all of the comments. We have replied to all your comments/suggestions, aiming to clarify our conceptual background, methodological procedures, and data analyses.

A point-by-point reply now follows. All substantive changes in the manuscript were highlighted in yellow. We greatly appreciate the opportunity to improve the manuscript based on these comments and suggestions. We are confident that the comments and the ensuing revisions made the manuscript a stronger contribution to the field.

  1. We thank for the comment and the opportunity to clarify the Transactional Model and its authors (please see p. 1, lines 40-61).
  2. According with the reviewer suggestion, we now specified the focus on preschool children to establish the need for the current review (please see p., x, lines 51-54).

  3. We thank the suggestion and we now provided a definition for sleep consolidation (please see p. x, lines 128-129).

  4. We appreciate the reviewer’s comment. Despite of having established quantitative studies as an inclusion criterium, in fact we did not find any qualitative or mixed study. Therefore, we eliminated this criterium.

  5. Following the reviewer’s suggestion, we added a column detailing not only the attachment measure, but also the attachment variables being measured. We also added the sleep variables measured.

Round 2

Reviewer 1 Report

I am fine with the authors' revision and I think that the manuscript is improved.